# Association Between Sarcopenia and Acupressure Testing in Older Adults Requiring Long-Term Care

**DOI:** 10.3390/medicina60111852

**Published:** 2024-11-11

**Authors:** Takahiro Shiba, Yohei Sawaya, Ryo Sato, Tamaki Hirose, Lu Yin, Masataka Shiraki, Masahiro Ishizaka, Akira Kubo, Tomohiko Urano

**Affiliations:** 1Integrated Facility for Medical and Long-Term Care, Care Facility for the Elderly “Maronie-en”, 533-4 Iguchi, Nasushiobara 329-2763, Tochigi, Japan; t-shiba@iuhw.ac.jp (T.S.); 1612010@g.iuhw.ac.jp (L.Y.); 2Nishinasuno General Home Care Center, Department of Day Rehabilitation, Care Facility for the Elderly “Maronie-en”, 533-11 Iguchi, Nasushiobara 329-2763, Tochigi, Japan; sawaya@iuhw.ac.jp (Y.S.); 19s1095@g.iuhw.ac.jp (R.S.); n-tamaki@iuhw.ac.jp (T.H.); 3Department of Physical Therapy, School of Health Sciences, International University of Health and Welfare, 2600-1 Kitakanemaru, Otawara 324-8501, Tochigi, Japan; ishizaka@iuhw.ac.jp; 4Research Institute and Practice for Involutional Diseases, 1610-1 Misatomeisei, Azumino 399-8101, Nagano, Japan; ripid@bh.wakwak.com; 5Department of Physical Therapy, Odawara School of Health Sciences, International University of Health and Welfare, 1-2-25 Shiroyama, Odawara 250-0045, Kanagawa, Japan; akubo@iuhw.ac.jp; 6Department of Geriatric Medicine, School of Medicine, International University of Health and Welfare, 4-3 Kozunomori, Narita 286-8686, Chiba, Japan

**Keywords:** acupressure testing, aged, grip strength, muscles, sarcopenia

## Abstract

*Background and Objectives*: This study analyzed the relationship between pressure pain test outcomes and sarcopenia in elderly patients and explored possible clinical applications. *Materials and Methods*: The participants included 143 older adults requiring long-term care who could be diagnosed with sarcopenia. Along with sarcopenia diagnosis, the participants underwent acupressure testing symmetrically at nine sites (occiput, lower cervical, trapezius, supraspinatus, second rib, lateral epicondyle, gluteus, greater trochanter, and knee), totaling 18 sites. The analyses included comparisons of sarcopenia status and total tender points between the groups and a multivariable analysis. The association between sarcopenia and the number of tender points were examined based on correlations between the number of tender points and grip strength, walking speed, and skeletal muscle mass index (SMI). Intergroup comparisons and multivariable analysis of tender points with and without sarcopenia were performed to investigate specific tender points associated with sarcopenia. *Results*: An independent association was observed between sarcopenia and the number of tender points (*p* = 0.001). Furthermore, the number of tender points was correlated with grip strength (ρ = −0.536, *p* < 0.001), walking speed (ρ = −0.200, *p* = 0.028), and SMI (ρ = −0.394, *p* < 0.001). The supraspinatus (*p* = 0.029, 95% confidence interval: 1.221–35.573) and lower cervical (*p* = 0.039, 95% confidence interval: 1.050–7.245) regions were identified as specific tender points. *Conclusions*: In older adult patients requiring long-term care, sarcopenia is associated with an increased number of tender points throughout the body, with the supraspinatus and lower cervical regions potentially being specific tender points. Acupressure testing for tenderness may be a useful assessment parameter in sarcopenia patients.

## 1. Introduction

Sarcopenia is defined as the loss of muscle mass and strength throughout the body caused by factors such as aging and disease [1], affecting at least 50 million people worldwide [2]. The definition of sarcopenia varies across regions and organizations, but in 2024, the Global Leadership Initiative in Sarcopenia (GLIS) reported a global conceptual definition, reaching the consensus that sarcopenia is a condition characterized by simultaneous reductions in muscle mass and muscle strength [3]. Sarcopenia has been associated with adverse outcomes such as fractures from falls [4], osteoporosis [5], depression [6], diabetes [7], liver dysfunction [8], increased hospitalization [9], and mortality [10]. Sarcopenia is considered the new geriatric giant among older adults [11]. As the global population ages, the number of patients with sarcopenia is expected to increase [12]. Moreover, the incidence of sarcopenia is particularly high among older adults who require support and nursing care [13], making the development of prevention and intervention strategies for sarcopenia an urgent issue.

Early detection to prevent sarcopenia is important, and many screening methods have been developed. Typical sarcopenia screening methods include calf circumference measurement and the SARC-F [14]. However, calf circumference is affected by edema [15], and the SARC-F has low sensitivity as a screening method [16]. Therefore, there is a need to develop a sarcopenia screening method that can be applied to older adults in need of care, even if they have diverse physical disabilities and varying levels of disability. We devised a screening method for sarcopenia on the basis of whether the cap of a plastic bottle can be opened [17] and the circumference of the neck [18].

Furthermore, as a new screening method for sarcopenia, we focused on a tenderness test that evaluates sensitivity to pain at each joint and muscle site [19,20,21]. Two main methods of testing tenderness exist, namely, using an algesiometer or acupressure. Quantitative evaluation can be performed via an algesiometer, which has been widely used in many tenderness threshold studies [19,21]. However, the acupressure method, which applies pressure to the affected area primarily with the examiner’s thumb, is widely used in general clinical practice because it does not require special equipment and can be performed anywhere [22].

Previous studies on tenderness have reported that muscle fatigue [23] and musculoskeletal disorders [24,25] lower the tenderness threshold. Furthermore, an association between musculoskeletal pain and sarcopenia [26,27] has been reported, indicating the role of central sensitization (CS) as a contributing factor [28]. CS is defined as the amplification of nerve signals that induce hyperalgesia in the central nervous system (brain and spinal cord) [29] and contributes to increased pain sensitivity and decreased tenderness thresholds in older adults [30]. When measured using an algesiometer, participants with sarcopenia had lower tenderness thresholds for the extensor digitorum longus and biceps brachii than participants without sarcopenia [28,31]. However, these studies were based on algesiometry. Previous studies have reported an association between tenderness and decreased balance and walking ability [32], but no studies have reported an association with sarcopenia.

Therefore, we investigated the relationship between tenderness testing and sarcopenia through acupressure in older adult patients requiring care and examined the potential for a new sarcopenia assessment index that could be implemented in clinical practice.

## 2. Materials and Methods

### 2.1. Research Design and Ethical Considerations

This cross-sectional study was conducted at a single institution. According to the guidelines of the Helsinki Convention, all participants received a full explanation of the study’s purpose and measurements and provided written informed consent. This study was approved by the Ethical Review Committee of the International University of Health and Welfare (Approval No. 17-lo-189-7, 20-B-445).

### 2.2. Participants

Figure 1 shows the participant selection process. The study involved 143 older adults who met the inclusion criteria of requiring support or care and participating in day-care rehabilitation between March 2021 and September 2022. Sarcopenia was diagnosed through body composition analysis and a gait test in the standing position. The exclusion criteria included individuals (i) who could not accurately understand the test due to dementia (*n* = 14), (ii) who could not answer the questionnaire accurately due to aphasia (*n* = 6), (iii) who could not obtain accurate results from body composition analysis because they were 100 years of age or older (*n* = 1), and (iv) whose acupressure test (*n* = 2) data were missing. Therefore, after 23 individuals were excluded, the remaining 120 participants (males, 75; females, 45; mean age, 77.9 ± 9.4 years) were included. No participants refused to participate.

### 2.3. Survey Content

#### 2.3.1. Diagnosis of Sarcopenia

The Asian Working Group for Sarcopenia 2019 (AWGS 2019) diagnostic criteria [33], which are widely used to determine sarcopenia in Asians, were used to diagnose sarcopenia. The presence of sarcopenia was determined by body composition analysis, gait tests, and grip strength tests on the basis of the AWGS 2019 algorithm.

Skeletal muscle mass index (SMI) was calculated by measuring limb skeletal muscle mass with the bioimpedance analysis (BIA) method using a body composition analyzer (InBody 520, InBody Japan, Inc., Seoul, Korea) and dividing by the square of height.

Grip strength was measured using a digital grip dynamometer (digital grip dynamometer Grip D-TKK5401, Takei Kiki Kogyo, Niigata, Japan). Grip strength was measured twice on each side (left and right) while the participant sat in a chair. The maximum value from the left and right grip measurements was used as the representative value.

The participants were asked to walk straight ahead at a comfortable walking speed on an 11 m path, consisting of a 5 m measurement section and a 3 m preliminary section at both ends of the measurement section. Walking speed was calculated from the time required for the measurement section. Measurements were taken twice, and the average walking speed was used as the representative value. Walking assistive devices were used for the participants.

Participants who had low skeletal muscle mass and either low muscle strength or low physical function were classified into the sarcopenia group, whereas those who did not meet the criteria for sarcopenia were classified into the nonsarcopenia group.

#### 2.3.2. Acupressure Test

Acupressure testing was performed using the tenderness sites [34] stipulated in the American College of Rheumatology classification criteria for fibromyalgia. Acupressure testing was performed symmetrically at nine acupressure sites (occiput, low cervical, trapezius, supraspinatus, second rib, lateral epicondyle, gluteus, greater trochanter, and knee), totaling 18 sites (Appendix A). All tests were conducted seated with the feet flat on the ground. Pressure was applied until the thumbnail of the dominant hand turned white (approximately 4 kg), following the methods outlined in the fibromyalgia diagnostic guidelines [35,36,37]. The acupressure test used in this study is reliable among raters [38,39] and is no less reliable than the test with a pressure pain meter [40]. The number of tender points and site-specific tender points were evaluated. A tender point was defined as tenderness on either the right, the left, or both sides of the nine acupressure sites. Acupressure testing was performed on all participants by a physical therapist with at least 10 years of experience. The average acupressure strength of the examiner was 4.08 ± 0.20 kgf, with a coefficient of variation of 4.9%.

#### 2.3.3. Basic Attributes

Age, sex, height, weight, body mass index (BMI), and level of care required were collected from the medical records of the participating institutions. The level of care required indicates the intensity of care needed for daily living, which depends on the participants’ levels of physical and mental functioning. It is classified into support levels 1 and 2 (levels requiring support) and levels 1 to 5 (levels requiring nursing care) [41]. In this study, support level 1, with the lowest care level, was classified as 1, whereas care level 5, with the highest care level, was classified as 7.

#### 2.3.4. Other Evaluation Items

Nutritional status was assessed using the Mini Nutritional Assessment Short-Form (MNA-sf). Nutritional status was evaluated using a 14-point scale, where higher scores indicate better nutritional status [42]. The SARC-F was also used to screen for sarcopenia. The SARC-F is a questionnaire consisting of five questions: strength (S), assistance walking (A), rising from a chair (R), climbing stairs (C), and falling (F), with a total score of 10. A score of 4 or more indicates the possibility of sarcopenia [14].

### 2.4. Analysis

Each survey item (age, sex, height, weight, BMI, level of care needed, grip strength, walking speed, SMI, MNA-sf, SARC-F, and total number of tender points) was compared between the sarcopenia and nonsarcopenia groups using unpaired *t* tests, Mann–Whitney U tests, and χ^2^ tests to investigate the associations between the number of tender points and sarcopenia. Next, we used Spearman’s rank correlation coefficient to investigate the correlation between the number of tender points and the diagnostic items of sarcopenia (grip strength, walking speed, and SMI), the degree of nursing care needed, MNA-sf, BMI, and SARC-F. Furthermore, a binomial logistic regression analysis (adjusted for age and sex) was performed with sarcopenia as the dependent variable and the total number of tender points as the independent variable to investigate whether sarcopenia was independently associated with the number of tender points.

Next, a χ^2^ test was performed on the 18 tender points throughout the body to investigate sarcopenia-specific tender points, comparing sites of tenderness on either the left or right side with sarcopenia status. A binomial logistic regression analysis (adjusted for age and sex) was then performed, with the dependent variable being the presence or absence of sarcopenia and the independent variable being tender points with a significant difference. SPSS Statistics version 22 (IBM, Tokyo, Japan) was used for the analysis, with the significance level set at *p* < 0.05.

### 2.5. Power Analysis

Post hoc analysis was conducted to evaluate statistical power. Using the Mann–Whitney U test model, the effect size for sarcopenia and tenderness points was 0.576, with an alpha error of 0.05 and a sample size of 68 in the sarcopenia group and 52 in the nonsarcopenia group, resulting in a power of 0.92.

## 3. Results

A comparison of the sarcopenia and nonsarcopenia groups revealed that the sarcopenia group was significantly older (*p* = 0.009); required a greater level of nursing care (*p* = 0.026); had more tender points (*p* = 0.001); and had significantly lower weight (*p* < 0.001), BMI (*p* < 0.001), grip strength (*p* < 0.001), walking speed (*p* < 0.001), SMI (*p* < 0.001), and MNA-sf (*p* < 0.001) (Table 1). Analysis of the tenderness sites revealed that the occiput (*p* = 0.031), low cervical (*p* = 0.003), supraspinatus (*p* = 0.007), and greater trochanter (*p* = 0.015) were significantly more tender in individuals with sarcopenia (Table 1).

The number of tender points was significantly correlated with all the diagnostic indicators of sarcopenia: grip strength (ρ = −0.536), walking speed (ρ = −0.200), and SMI (ρ = −0.394). The number of tender points also correlated with MNA-sf (ρ = −0.333) and the level of nursing care required (ρ = 0.181). However, the SARC-F—a screening tool for sarcopenia—showed significant differences only in gait speed (ρ = −0.461) and MNA-sf (ρ = −0.274) (Table 2). The correlations between the number of tender points and each survey item by sex are shown in Appendix A.

A binomial logistic regression analysis (adjusted variables: age and sex), in which sarcopenia status was the dependent variable and the number of tender points was the independent variable, revealed an independent association (*p* < 0.001, odds ratio (OR): 1.296) between sarcopenia and the number of tender points (Table 3). Furthermore, in a binomial logistic regression analysis (adjusted variables: age and sex), where sarcopenia status was the dependent variable and the site of tenderness was the independent variable, the lower cervical (*p* = 0.039, OR: 2.758) and supraspinatus (*p* = 0.029, OR: 6.773) regions were independently associated (Table 4).

## 4. Discussion

This study is the first to use acupressure tests to investigate sarcopenia and whole-body tenderness. These results revealed that (i) the number of tender points throughout the body increased in participants with sarcopenia, (ii) the number of tender points correlated with the items used in the diagnostic criteria for sarcopenia (grip strength, walking speed, and SMI), and (iii) the supraspinatus and lower cervical regions were tender points independently associated with sarcopenia.

Unlike conventional screening methods for sarcopenia, the acupressure tenderness test can be performed without equipment. Additionally, it does not require a large space and is commonly performed by physicians and physical therapists in their usual practice. In a previous study that investigated the reliability of acupressure tenderness testing using a similar method to ours, the test was performed on 15–30 participants with two examiners, and moderate to high agreement among the examiners was observed [38,39]. Moreover, acupressure testing is as reliable as testing with a pressure gauge [40]. Therefore, a tenderness test using acupressure may be helpful in identifying sarcopenia.

Previous studies on tenderness thresholds, muscle strength, and physical function have reported that lower tenderness thresholds in the deltoid and short radial carpal extensor muscles in older adults are associated with low grip strength [43]. Knee extension strength was related to tenderness thresholds for isokinetic and isometric contractions in people with fibromyalgia [44]. In a study of 585 community-dwelling older adults, Laura et al. reported that those with increased or widespread tenderness had slower walking speeds and a low short physical performance battery [32]. Furthermore, a previous study examining tenderness thresholds and skeletal muscle mass reported that tenderness thresholds were low in individuals with high body fat and low skeletal muscle mass [45], indicating that a decrease in the cross-sectional area of lumbar muscles (multifidus and erector spinae) may increase sensitivity to tenderness [46]. These reports support the association between sarcopenia and tenderness observed in this study. However, many previous studies used pressure gauges rather than acupressure. This suggests that similar evaluations could be conducted without using specific tools. The number of tender points throughout the body can be a form of multifaceted screening, as it has been correlated with sarcopenia, the degree of nursing care needed, and the nutritional index MNA-sf.

Furthermore, a decreased tenderness threshold has also been associated with systemic inflammation. A study of 27 healthy adult volunteers treated with purified *Escherichia coli* endotoxin reported a 20.0 ± 4.0% decrease in the tenderness threshold compared with that of the controls, along with increases in interleukin-6 (IL-6) and tumor necrosis factor-alpha (TNF-α) after administration [47]. Moreover, IL-6 levels are also greater in patients with sarcopenia than in those without [48]. Our findings align with those of previous studies, suggesting that sarcopenia is associated with increased tender points throughout the body, which may indicate a state of sarcopenia.

The present study revealed that tenderness in the lower cervical and supraspinatus regions was associated with sarcopenia. Few reports have examined the association between tenderness and sarcopenia; therefore, more detailed biological and biochemical investigations are needed to clarify the mechanism of this association in the lower cervical region and supraspinatus. However, these results suggest that sarcopenia screening can be further simplified by examining tenderness only in the lower cervical and supraspinatus regions without needing to examine tenderness in nine locations. In the future, longitudinal studies should be conducted to determine whether the development and progression of sarcopenia in patients with tenderness in the lower cervical and supraspinatus regions are involved in long-term prognosis, such as a decrease in activities of daily living.

This study has several limitations. First, this is a single-center study and facility bias is expected. Therefore, similar studies need to be conducted in other facilities to adapt this study to the entire older adult population needing support or care. Second, we could not confirm whether the IL-6 and TNF-α levels were elevated because blood tests were not performed. Third, we did not investigate the relationships between sarcopenia and body fat percentage; therefore, we could not confirm whether the patients had sarcopenic obesity. Fourth, this is a cross-sectional study, and a longitudinal study is needed to explain the causal relationship between sarcopenia and increased tender points. Therefore, future research should examine the relationships of sarcopenia in more detail by analyzing acupressure test results and verifying the results with blood biochemistry tests and body fat percentage. Sarcopenia obesity has been reported to be particularly associated with high mortality [49] but is often not noted in screening for sarcopenia. Future studies should, therefore, examine whether acupressure testing can affect sarcopenia obesity. However, the acupressure test, which can be performed without specific tools or venue restrictions, may be a clinically applicable method for assessing sarcopenia in various settings, such as during bedside evaluations or home visits.

## 5. Conclusions

In older adults requiring long-term care, sarcopenia has been associated with an increase in tender points throughout the body. In particular, the supraspinatus and lower cervical spine regions were shown to be specific tender points, and it may be possible to evaluate sarcopenia only by tenderness testing at specific points. In older adults requiring long-term care with a variety of disabilities, the acupressure test, which can be performed without equipment, may be a useful index for evaluating sarcopenia.

## Figures and Tables

**Figure 1 medicina-60-01852-f001:**
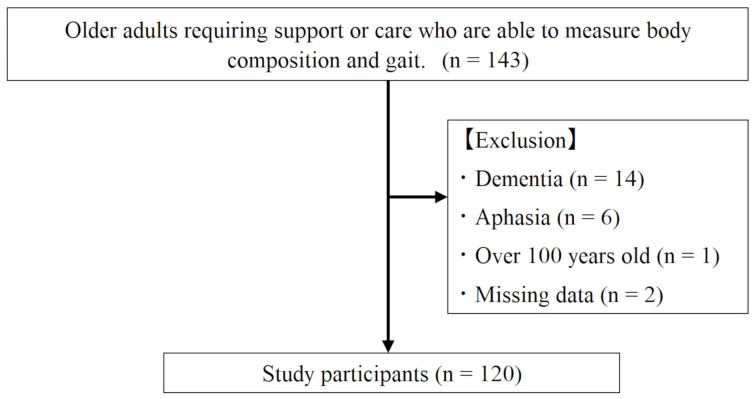
Participant selection flow.

**Table 1 medicina-60-01852-t001:** Comparison of sarcopenia and nonsarcopenia.

	Sarcopenia (*n* = 68)	Nonsarcopenia (*n* = 52)	*p*
Age (years)	79.9 ± 8.6	75.4 ± 9.9	0.009 *
Female, *n* (%)	22 (32.4)	23 (44.2)	0.183
Height (cm)	158.1 ± 8.3	161.1 ± 9.3	0.066
Weight (kg)	54.4 ± 7.4	62.6 ± 11.9	<0.001 *
BMI (kg/m^2^)	21.8 ± 2.8	24.1 ± 4.1	<0.001 *
Care levels (1–7) ^†^	3.0 (2.0–4.0)	2.0 (1.0–3.0)	0.026 *
Grip strength (kg)	20.3 ± 6.4	26.2 ± 9.2	<0.001 *
Gait speed (m/sec)	0.7 ± 0.3	0.9 ± 0.3	<0.001 *
SMI (kg/m^2^)	5.9 ± 0.7	7.2 ± 0.9	<0.001 *
MNA-sf (score) ^†^	11.0 (9.0–13.0)	13.0 (12.0–14.0)	<0.001 *
SARC-F (Score) ^††^	4.0 (2.0–7.0)	3.0 (1.0–5.5)	0.207
Total number of tenderness (score) ^†^	5.0 (2.0–7.0)	2.0 (0.0–4.0)	0.001 *
Occiput, *n* (%)	17 (25.0)	5 (9.6)	0.031 *
Low cervical, *n* (%)	55 (80.9)	29 (55.8)	0.003 *
Trapezius, *n* (%)	20 (29.4)	11 (21.2)	0.306
Supraspinatus, *n* (%)	14 (20.6)	2 (3.8)	0.007 *
Second rib, *n* (%)	34 (50.0)	20 (38.5)	0.208
Lateral epicondyle, *n* (%)	4 (5.9)	1 (1.9)	0.387
Gluteus, *n* (%)	30 (44.1)	17 (32.7)	0.204
Greater trochanter, *n* (%)	24 (35.3)	8 (15.4)	0.015 *
Knee, *n* (%)	24 (35.3)	14 (26.9)	0.329

* *p* < 0.05. ^†^: Median (25th–75th percentile). ^††^: With missing data (73 men, 45 women). BMI, Body mass index; SMI, skeletal muscle mass index; MNA-sf, Mini Nutritional Assessment. *T* test: age, grip strength, gait speed, and SMI. Mann–Whitney U test: care levels, MNA-sf, total number of tenderness. χ^2^ test: Female.

**Table 2 medicina-60-01852-t002:** Correlation between the number of tender points and sarcopenia diagnostic items.

	Number of Tender Points	SARC-F ^†^	Grip Strength	Gait Speed	SMI	BMI	MNA-sf	Care Levels
Total number of tender points	-							
SARC-F ^†^	0.166	-						
Grip strength	−0.536 **	−0.111	-					
Gait speed	−0.200 *	−0.461 **	0.174	-				
SMI	−0.394 **	−0.079	0.650 **	0.216 *	-			
BMI	−0.121	−0.063	0.093	0.147	0.374 **	-		
MNA-sf	−0.333 **	−0.274 **	0.323 **	0.144	0.371 **	0.375 **	-	
Care levels	0.181 *	0.165	−0.148	−0.325 **	−0.191 *	−0.195 *	−0.213 *	-

* *p* < 0.05, ** *p* < 0.01. ^†^: With missing data (73 men, 45 women). SMI, skeletal muscle mass index; BMI, body mass index; MNA-sf, Mini Nutritional Assessment Short-Form.

**Table 3 medicina-60-01852-t003:** Binomial logistic regression analysis of sarcopenia status and the number of tender points.

	β	SE	*p*	OR	95%CI
Number of tender points	0.259	0.073	<0.001 *	1.296	1.122–1.497

* *p* < 0.05 Adjusted: Age, sex. SE, standard error; OR, odds ratio; CI, confidence interval. Dependent variables: Sarcopenia = 1, Nonsarcopenia = 0. Independent variables: Number of tender points.

**Table 4 medicina-60-01852-t004:** Binomial logistic regression analysis of sarcopenia status and tenderness sites.

	β	SE	*p*	OR	95% CI
Occiput	0.540	0.653	0.408	1.717	0.477–6.174
Supraspinatus	1.913	0.874	0.029 *	6.773	1.221–37.573
Greater trochanter	0.555	0.558	0.321	1.741	0.583–5.203
Low cervical	1.015	0.493	0.039 *	2.758	1.050–7.245

* *p* < 0.05 Adjusted: age, sex. SE, standard error; OR, odds ratio; CI, confidence interval. Dependent variables: Sarcopenia = 1, Nonsarcopenia = 0. Independent variables: With tenderness = 1, no tenderness = 0.

## Data Availability

The raw measurements are available in the Appendix A.

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
