# Peer review of "Association Between Sarcopenia and Acupressure Testing in Older Adults Requiring Long-Term Care"

_medicina, 2024, doi:10.3390/medicina60111852_

Round 1

Reviewer 1 Report

Comments and Suggestions for Authors

Dear Authors, 

The topic you have presented demonstrates a significant level of originality and interest, and I congratulate you on this. Please allow me to make a few observations that may prove helpful to you.

In the section Discussion:

1)There is No mention of inter-rater reliability: While the qualifications of the physical therapist are stated, no details are given about how inter-rater variability was controlled, especially since acupressure testing can vary depending on the examiner’s technique.

2)There is No discussion on study limitations: The limitations mentioned are valid but not sufficiently explored. For example, the authors mention that the study is single-center, but do not discuss the potential implications of this for generalizability. Additionally, the absence of a control group or longitudinal data is not mentioned

3)There is No mention of implications for clinical practice: Although the discussion presents interesting findings, the paper would benefit from more practical recommendations for clinicians on how to integrate acupressure testing into routine sarcopenia screening, especially in resource-limited settings.

Thank you

Author Response

添付ファイルをご覧ください。

Reviewer 2 Report

Comments and Suggestions for Authors

Thank you for submitting an interesting research paper suggesting acupressure testing as a method for diagnosing sarcopenia. Please edit the instructions below to make it more meaningful.

Abstract

1. Please write specifically so that the purpose of the study can be conveyed more clearly. To this end, please explain the purpose and research method simply and clearly. (Example: This study analyzed the relationship between the pressure pain test and sarcopenia in elderly patients and explored the possibility of clinical application.)

2. Please add specific statistical results to the results. Please use p-values and/or confidence intervals, etc. to indicate them.

Intro.

1. The introduction section lacks description of why acupressure should be used as a diagnostic method. Please describe in detail and specifically why the acupressure pain test was selected as a tool for assessing sarcopenia.

2. The introduction section requires description of the definition and influence of sarcopenia, and its relationship to acupressure in previous studies. Please describe in detail.

3. More references are needed to compare methods and tools for evaluating the acupressure test. Please clearly describe how this study is different from previous studies and add other references to describe the differences.

Method

1. Please add the reason for determining the number of study subjects and the method for calculating the sample size.

2. Please describe in detail the method of selecting the study subjects.

3. Please describe in detail the method of random assignment of the study subjects or the sample selection procedure.

4. Please describe in detail and scientifically the method of conducting the acupressure test.

5. Please describe in more detail and clearly the selection and exclusion criteria of the study subjects.

6. Please specify in detail the method of handling missing data.

Result

 1. Please clearly describe the number of dropouts and the reasons for them.

Discussion

1. Please describe in more detail the comparison with previous studies and discuss further the differences of this study.

2. Please explain the limitations of the study in more detail.

3. Please describe in detail how this study can be applied in clinical practice.

4. The results of the study showed that pressure pain in the supraspinatus and neck areas was associated with sarcopenia. Please emphasize this more in the conclusion.

Round 2

Reviewer 2 Report

Comments and Suggestions for Authors

I have carefully reviewed the revised manuscript. It appears to have been appropriately revised in accordance with the reviewer's comments. Thank you.